# Controlling Neural Networks with Rule Representations

**Sungyong Seo, Sercan Ö. Arık, Jinsung Yoon, Xiang Zhang, Kihyuk Sohn, Tomas Pfister**
Google Cloud AI
Sunnyvale, CA, USA
`{sungyongs,soarik,jinsungyoon,fancyzhx,kihyuks,tpfister}@google.com`

## Abstract

We propose a novel training method that integrates rules into deep learning, in a way the strengths of the rules are controllable at inference. Deep Neural Networks with Controllable Rule Representations (DEEPCTRL) incorporates a rule encoder into the model coupled with a rule-based objective, enabling a shared representation for decision making. DEEPCTRL is agnostic to data type and model architecture. It can be applied to any kind of rule defined for inputs and outputs. The key aspect of DEEPCTRL is that it does not require retraining to adapt the rule strength – at inference, the user can adjust it based on the desired operation point on accuracy vs. rule verification ratio. In real-world domains where incorporating rules is critical – such as Physics, Retail and Healthcare – we show the effectiveness of DEEPCTRL in teaching rules for deep learning. DEEPCTRL improves the trust and reliability of the trained models by significantly increasing their rule verification ratio, while also providing accuracy gains at downstream tasks. Additionally, DEEPCTRL enables novel use cases such as hypothesis testing of the rules on data samples, and unsupervised adaptation based on shared rules between datasets.

## 1 Introduction

Deep neural networks (DNNs) excel at numerous tasks such as image classification [28, 29], machine translation [22, 30], time series forecasting [11, 21], and tabular learning [2, 25]. DNNs get more accurate as the size and coverage of training data increase [17]. While investing in high-quality and large-scale labeled data is one path, another is utilizing prior knowledge – concisely referred to as 'rules': reasoning heuristics, equations, associative logic, constraints or blacklists. In most scenarios, labeled datasets are not sufficient to teach all rules present about a task [4, 12, 23, 24]. Let us consider an example from Physics: the task of predicting the next state in a double pendulum system, visualized in Fig. 1. Although a 'data-driven' black-box model, fitted with conventional supervised learning, can fit a relatively accurate mapping from the current state to next, it can easily fail to capture the canonical rule of 'energy conservation'. In this work, we focus on how to teach 'rules' in effective ways so that DNNs absorb the knowledge from them in addition to learning from the data for the downstream task.

The benefits of learning from rules are multifaceted. First, rules can provide extra information for cases with minimal data supervision, improving the test accuracy. Second, the rules can improve trust and reliability of DNNs. One major bottleneck for widespread use of DNNs is them being 'black-box'. The lack of understanding of the rationale behind their reasoning and inconsistencies of their outputs with human judgement often reduce the trust of the users [3, 26]. By incorporating rules, such inconsistencies can be minimized and the users' trust can be improved. For example, if a DNN for loan delinquency prediction can absorb all the decision heuristics used at a bank, the loan officers of the bank can rely on the predictions more comfortably. Third, DNNs are sensitive to slight changes

35th Conference on Neural Information Processing Systems (NeurIPS 2021).

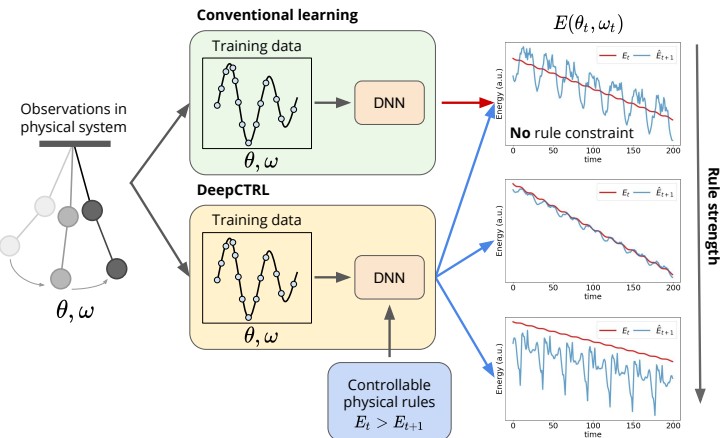

Figure 1: Overview of DEEPCTRL. When a DNN is trained only with the task-specific objective (in this example, predicting next positions/velocities of two objects connected in a pendulum), it may easily violate the rule ($E_t > E_{t+1}$) that it must have followed according to the energy damping rule from physics (top graph). DEEPCTRL (with outputs shown via blue arrows) provides a controllable mechanism that enables the rule dependency to be adjusted at inference time in order to achieve an optimal behavior (middle graph) in regards to accuracy and rule verification ratio. With increased rule strength, DEEPCTRL yields an operation point (bottom graph) where is satisfied for all time steps.

to the inputs that are human-imperceptible [15, 20, 31]. With rules, the impact of these changes can be minimized as the model search space is further constrained to reduce underspecification [7, 10].

When 'data-driven' and 'rule-driven' learning are considered jointly, a fundamental question is how to balance the contribution from each. Even when a rule is known to hold 100% of the time (such as the principles in natural sciences), the contribution of rule-driven learning should not be increased arbitrarily. There is an optimal trade-off that depends not only on the dataset, but also on each sample. If there are training samples that are very similar to a particular test sample, a weaker rule-driven contribution would be desirable at inference. On the other hand, if the rule is known to hold for only a subset of samples (e.g. in Retail, the varying impact of a price change on different products [6]), the strength of the rule-driven learning contribution should reflect that. Thus, a framework where the contributions of data- and rule-driven learning can be controlled would be valuable. Ideally, such control should be enabled at inference without the need for retraining in order to minimize the computational cost, shorten the deployment time, and to adjust to different samples or changing distributions flexibly.

In this paper, we propose DEEPCTRL that enables joint learning from labeled data and rules. DEEPCTRL employs separate encoders for data and rules with the outputs combined stochastically to cover intermediate representations coupling with corresponding objectives. This representation learning is the key to controllability, as it allows increasing/decreasing the rule strength gradually at inference without retraining. To convert any non-differentiable rules into differentiable objectives, we propose a novel perturbation-based method. DEEPCTRL is agnostic to the data type or the model architecture, and DEEPCTRL can be flexibly used in different tasks and with different rules. We demonstrate DEEPCTRL on important use cases from Physics, Retail, and Healthcare, and show that it: (i) improves the rule verification ratio significantly while yielding better accuracy by merely changing the rule strength at inference; (ii) enables hypotheses to be examined for each sample based on the optimal ratio of rule strength without retraining (for the first time in literature, to the best of our knowledge); and (iii) improves target task performance by changing the rule strength, a desired capability when different subsets of the data are known to satisfy rules with different strengths.

**Related Work:** Various methods have been studied to incorporate 'rules' into deep learning, considering prior knowledge in wide range of applications. Posterior regularization [13] is one approach to inject rules into predictions. In [18], a teacher-student framework is used, where the teacher network is obtained by projecting the student network to a (logic) rule-regularized subspace and the student network is updated to balance between emulating the teacher's output and predicting the labels. Adversarial learning is utilized in [32], specifically for bias rules, to penalize unwanted biases. In [12]

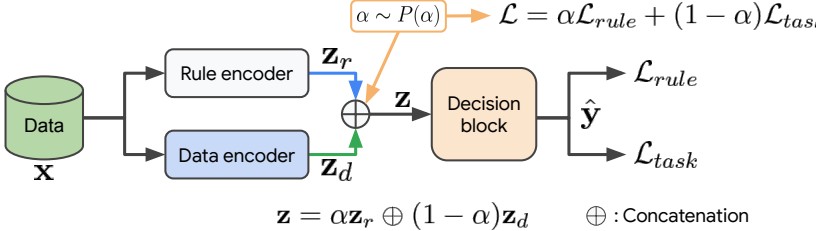

$$\mathbf{z} = \alpha\mathbf{z}_r \oplus (1-\alpha)\mathbf{z}_d \qquad \oplus : \text{Concatenation}$$

Figure 2: DEEPCTRL for controllable incorporation of a rule within the learning process. DEEPC-TRL introduces two passages for the input-output relationship, a data encoder and rule encoder, that produce two latent representations $z_r$ and $z_d$. These representations are stochastically concatenated with a control parameter $\alpha$ into a single representation $z$. $z$ is then fed into a decision block with objectives for each representations, $\mathcal{L}_{rule}$ and $\mathcal{L}_{task}$, again weighed by the control parameter $\alpha$. $\alpha$ is randomly sampled during training and set by users at inference to adjust the rule strength.

a framework is proposed that exploits Lagrangian duality to train with rules. In [24] learning with constraints is studied, via formulation over the space of confusion matrices and optimization solvers that operate through a sequence of linear minimization steps. In [27] constraints are injected via KL divergence for variational autoencoders, for controlling output diversity or disentangled latent factor representations. DEEPCTRL differentiate from the all aforementioned in how it injects the rules, with the aspect that it allows controllability of the rule strength at the inference without retraining, enabled by accurate learning of the rule representations in the data manifold. This unlocks new capabilities, beyond simply improving rule verification for a target accuracy.

## 2    Learning Jointly from Rules and Task

Let us consider the conventional approach [9, 12, 14] for incorporating rules via combining the training objectives for the supervised task and a term that denotes the violation of rule:

$$\mathcal{L} = \mathcal{L}_{task} + \lambda\mathcal{L}_{rule}, \tag{1}$$

where $\lambda$ is the coefficient for the rule-based objective. There are three limitations of this approach that we aim to address: (i) $\lambda$ needs to be defined before learning (e.g. can be a hyperparameter guided with validation score), (ii) $\lambda$ is not adaptable to target data at inference if there is any mismatch with the training setup, and (iii) $\mathcal{L}_{rule}$ needs to be differentiable with respect to learnable parameters.

DEEPCTRL modifies canonical training by creating rule representations, coupled with data representations, which is the key to enable the rule strength to be controlled at inference time. Fig. 2 overviews DEEPCTRL and Algorithm 1 describes its corresponding training process. We propose to modify the canonical training approach by introducing *two passages* for the input-output relationship of the task, via *data encoder* $\phi_d$ and *rule encoder* $\phi_r$. In this way, our goal is to make each encoder individually learn the latent representations ($z_d$ and $z_r$), corresponding to extracted information from the labeled data and the rule. Then, the two representations are stochastically concatenated (with the operation denoted as $\oplus$) to obtain a single representation $z$. To adjust the relative contributions of data vs. rule encoding, we use a random variable $\alpha \in [0, 1]$, which also couples $(z_d, z_r)$ with the corresponding objectives $(\mathcal{L}_{task}, \mathcal{L}_{rule})$ (Lines 4 & 5 in Algorithm 1). $\alpha$ is sampled from the distribution $P(\alpha)$. The motivation to use a random $\alpha$ is to encourage learning the mapping with a range of values, so that at inference, the model can yield high performance with any particular chosen value. The output of the decision block ($\hat{y}$) is used in the entire objective.

By modifying the control parameter $\alpha$ at inference, users can control the behavior of the model to adapt it to unseen data. The strength of the rule on the output decision can be enhanced by increasing the value of $\alpha$. Setting $\alpha = 0$ would minimize the contribution of the rule at inference, but as shown in the experiments, the result can still be better than in conventional training since during the training process a wide range of $\alpha$ are considered. Typically, an intermediate $\alpha$ value yields the optimal solution given specific performance, transparency and reliability goals. To ensure that a model shows distinct and robust behavior when $\alpha \to 0$ or $\alpha \to 1$ and thus interpolates accurately later, we propose to sample $\alpha$ more heavily at the two extremes rather than uniformly from $[0, 1]$. To this end, we choose to sample from a Beta distribution (Beta$(\beta, \beta)$). We observe strong results with $\beta = 0.1$ in

---

**Algorithm 1** Training process for DEEPCTRL.

---

**Input**: Training data $\mathcal{D} = \{(\boldsymbol{x}_i, \boldsymbol{y}_i) : i = 1, \cdots, N\}$.
**Output**: Optimized parameters
**Require**: Rule encoder $\phi_r$, data encoder $\phi_d$, decision block $\phi$, and distribution $P(\alpha)$.

1: Initialize $\phi_r, \phi_d$, and $\phi$
2: **while** not converged **do**
3:     Get mini-batch $\mathcal{D}_b$ from $\mathcal{D}$ and $\alpha_b \in \mathbb{R}$ from $P(\alpha)$
4:     Get $\boldsymbol{z} = \alpha_b \boldsymbol{z}_r \oplus (1 - \alpha_b \boldsymbol{z}_d)$ where $\boldsymbol{z}_r = \phi_r(\mathcal{D}_b)$ and $\boldsymbol{z}_d = \phi_d(\mathcal{D}_b)$.
5:     Get $\hat{\boldsymbol{y}} = \phi(\boldsymbol{z})$ and compute $\mathcal{L} = E_{\alpha \sim P(\alpha)}[\alpha \mathcal{L}_{rule} + \rho(1 - \alpha)\mathcal{L}_{task}]$ where $\rho = \mathcal{L}_{rule,0}/\mathcal{L}_{task,0}$
6:     Update $\phi_r, \phi_d$, and $\phi$ from gradients $\nabla_{\phi_r}\mathcal{L}, \nabla_{\phi_d}\mathcal{L}$, and $\nabla_\phi \mathcal{L}$
7: **end while**

---

most cases and in Section 5, we further study the impact of the selection of the prior for $\alpha$. Since a Beta distribution is highly polarized, each encoder is encouraged to learn distinct representations associated with the corresponding encoder rather than mixed representations. Similar sampling ideas were also considered in [1, 33] to effectively sample the mixing weights for representation learning.

One concern in the superposition of $\mathcal{L}_{task}$ and $\mathcal{L}_{rule}$ is their scale differences that may cause all learnable parameters to be dominated by one particular objective regardless of $\alpha$, and hence become unbalanced. This is not a desired behavior as it significantly limits the expressiveness of DEEPCTRL and may cause convergence into a single mode. E.g., if $\mathcal{L}_{rule}$ is much larger than $\mathcal{L}_{task}$, then DEEPCTRL will become a rule-based model even when $\alpha$ is close to 0. To minimize such imbalanced behavior, we propose to adjust the scaling automatically. Before starting a learning process, we compute the initial loss values $\mathcal{L}_{rule,0}$ and $\mathcal{L}_{task,0}$ on a training set and introduce a scale parameter $\rho = \mathcal{L}_{rule,0}/\mathcal{L}_{task,0}$. Overall, the DEEPCTRL objective function becomes:

$$\mathcal{L} = E_{\alpha \sim P(\alpha)}[\alpha \mathcal{L}_{rule} + \rho(1 - \alpha)\mathcal{L}_{task}]. \tag{2}$$

DEEPCTRL is model-type agnostic – we can choose and appropriate inductive bias for encoders and the decision block based on the type and task. E.g. if the input data is an image and the task is classification, the encoders can be convolutional layers to extract hidden representations associated with local spatial coherence, and the decision block can be an MLP followed by a softmax layer.

## 3  Integrating Rules via Input Perturbations

Algorithm 1 requires a rule-based objective $\mathcal{L}_{rule}$ that is a function of $(\boldsymbol{x}, \hat{\boldsymbol{y}})$ and is only differentiable with respect to the learnable parameters of the model. In some cases, it is straightforward to convert a rule into a differentiable form. For example, for a rule defined as $r(\boldsymbol{x}, \hat{\boldsymbol{y}}) \leq \tau$ given a differentiable function $r(\cdot)$, we can propose $\mathcal{L}_{rule} = \max(r(\boldsymbol{x}, \hat{\boldsymbol{y}}) - \tau, 0)$ that has a penalty with an increasing amount as the violation increases. However, there are many valuable rules that are non-differentiable with respect to the input $\boldsymbol{x}$ or learnable parameters, and in these cases it may not be possible to define a continuous function $\mathcal{L}_{rule}$ as above. Some examples include expressive statements represented as concatenations of Boolean rules (e.g. fitted decision trees) [8], such as *'The probability of the j-th class $\hat{\boldsymbol{y}}_j$ is higher when $a < \boldsymbol{x}_k$ (where a is a constant and $\boldsymbol{x}_k$ is the k-th feature)'* or *'The number of sales is increased when price of an item is decreased.'*.

We introduce an input perturbation method, overviewed in Algorithm 2, to generalize DEEPCTRL to non-differentiable constraints. The method is based on introducing a small perturbation $\delta \boldsymbol{x}$ (Line 4 in Algorithm 2) to input features $\boldsymbol{x}$ in order to modify the original output $\hat{\boldsymbol{y}}$ and construct the rule-based constraint $\mathcal{L}_{rule}$ for it. For example, if we were to incorporate the first sample rule in the previous paragraph (concatenations of Boolean rules), we would only consider $\boldsymbol{x}_p$ as a valid perturbed input when $\boldsymbol{x}_k < a$ and $a < \boldsymbol{x}_{p,k}$ and $\hat{\boldsymbol{y}}_p$ is computed from the $\boldsymbol{x}_p$. $\mathcal{L}_{rule}$ is defined as:

$$\mathcal{L}_{rule}(\boldsymbol{x}, \boldsymbol{x}_p, \hat{\boldsymbol{y}}_j, \hat{\boldsymbol{y}}_{p,j}) = \text{ReLU}(\hat{\boldsymbol{y}}_j - \hat{\boldsymbol{y}}_{p,j}) \cdot I(\boldsymbol{x}_k < a) \cdot I(\boldsymbol{x}_{p,k} > a). \tag{3}$$

This perturbation-based loss function is combined with a task-based loss function (Line 6 in Algorithm 2) to update all parameters in $\phi_r, \phi_d$, and $\phi$. The control parameter $\alpha$ is used for both $\hat{\boldsymbol{y}}$ and $\hat{\boldsymbol{y}}_p$, and $\mathcal{L}_{task}$ only considers the original output $\hat{\boldsymbol{y}}$. All learnable parameters in the rule encoder, data

---

**Algorithm 2** DEEPCTRL via perturbation-based integration of rules.

---

**Input**: Training data $\mathcal{D} = \{(\boldsymbol{x}_i, \boldsymbol{y}_i) : i = 1, \cdots, N\}$.
**Output**: Optimized parameters
**Require**: Rule encoder $\phi_r$, data encoder $\phi_d$, decision block $\phi$, and distribution $P(\alpha)$.

1: Initialize $\phi_r, \phi_d$, and $\phi$
2: **while** not converged **do**
3:     Get mini-batch $\mathcal{D}_b$ from $\mathcal{D}$ and $\alpha_b \in \mathbb{R}$ from $P(\alpha)$
4:     Get perturbed input $\boldsymbol{x}_p = \boldsymbol{x} + \delta\boldsymbol{x}$ where $\boldsymbol{x} \in \mathcal{D}_b$
5:     Get $\boldsymbol{y}$ and $\boldsymbol{y}_p$ from $\boldsymbol{x}$ and $\boldsymbol{x}_p$ through $\phi_r, \phi_d, \phi$, and $\alpha_b$, respectively
6:     Define $\mathcal{L}_{rule} = \mathcal{L}_{rule}(\boldsymbol{x}, \boldsymbol{x}_p, \hat{\boldsymbol{y}}, \hat{\boldsymbol{y}}_p)$ based on a rule and $\mathcal{L}_{task} = \mathcal{L}_{task}(\boldsymbol{y}, \hat{\boldsymbol{y}})$ to compute
    $\mathcal{L} = \alpha_b \mathcal{L}_{rule} + (1 - \alpha_b)\mathcal{L}_{task}$
7:     Update $\phi_r, \phi_d$, and $\phi$ from gradient $\nabla_{\phi_r}\mathcal{L}, \nabla_{\phi_d}\mathcal{L}$, and $\nabla_{\phi}\mathcal{L}$
8: **end while**

---

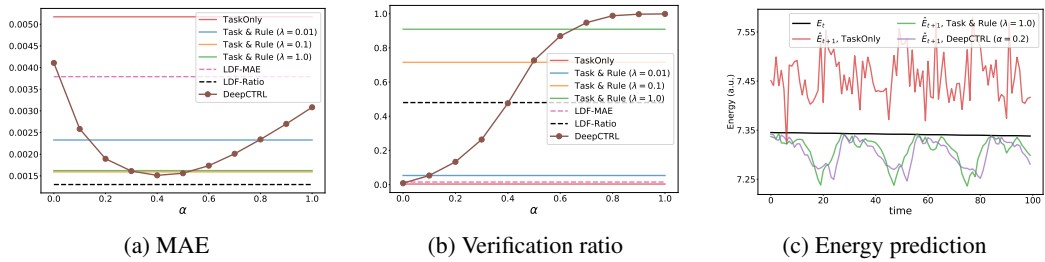

(a) MAE        (b) Verification ratio        (c) Energy prediction

Figure 3: Experimental results for the double pendulum task. (Left) Task-based prediction error and (Middle) verification ratio from different models. DEEPCTRL has a scale parameter which adjusts the scale mismatch between $\mathcal{L}_{task}$ and $\mathcal{L}_{rule}$. The performances of LDF method are highly sensitive to hyperparameter and its output is not reliable. (Right) Current and predicted energy at time $t$ and $t + 1$, respectively. According to the energy damping rule, $\hat{E}_{t+1}$ should not be larger than $E_t$.

encoder, and decision block are shared across the original input $\boldsymbol{x}$ and the perturbed input $\boldsymbol{x}_p$. Overall, this perturbation-based method expands the scope of rules we can incorporate into DEEPCTRL.

## 4 Experimental Results

We evaluate DEEPCTRL on machine learning use cases from Physics, Retail, and Healthcare, where utilization of rules is particularly important.

For the rule encoder ($\phi_r$), data encoder ($\phi_d$), and decision block ($\phi$), we use MLPs with ReLU activation at intermediate layers, similarly to [5, 16]. We compare DEEPCTRL to the TASKONLY baseline, which is trained with fixed $\alpha = 0$, i.e. it only uses a data encoder ($\phi_d$) and a decision block ($\phi$) to predict a next state. In addition, we include TASKONLY with rule regularization, TASK&RULE, based on Eq. 1 with a fixed $\lambda$. We make a comparison to Lagrangian Dual Framework (LDF) [12] that enforces rules by solving a constraint optimization problem.

### 4.1 Improved Reliability Given Known Principles

By adjusting the control parameter $\alpha$, a higher rule verification ratio, and thus more reliable predictions, can be achieved. Operating at a better verification ratio could be beneficial for performance, especially if the rules are known to be (almost) always valid as in natural sciences. We demonstrate a systematic framework to encode such principles from known literature, although they may not be completely learnable from training data.

**Dataset, task and the rule:** Following [4], we consider the time-series data generated from double pendulum dynamics with friction, from a given initial state $(\theta_1, \omega_1, \theta_2, \omega_2)$ where $\theta, \omega$ are angular displacement and velocity, respectively. We first generate the time-series data with a high sampling frequency (200Hz) to avoid numerical errors, and then downsample to a lower sampling frequency (10Hz) to construct the training dataset. We define the task as predicting the next state $\boldsymbol{x}_{t+1}$ of the

double pendulum from the current state $\boldsymbol{x}_t = (\theta_{1t}, \omega_{1t}, \theta_{2t}, \omega_{2t})$. We construct a synthetic training dataset using the analytically-defined relationship between inputs and outputs, and introduce a small additive noise ($\epsilon \sim \mathcal{N}(0, 0.0001)$) to model measurement noise in the setup. We focus on teaching the rule of energy conservation law. Since the system has friction, $E(\boldsymbol{x}_t) > E(\boldsymbol{x}_{t+1})$, where $E(\boldsymbol{x})$ is the energy of a given state $\boldsymbol{x}$. We apply an additional constraint based on the law such that $\mathcal{L}_{rule}(\boldsymbol{x}, \hat{\boldsymbol{y}}) = \text{ReLU}(E(\hat{\boldsymbol{y}}) - E(\boldsymbol{x}))$. To quantify how much the rule is learned, we evaluate the 'verification ratio', defined as the ratio of samples that satisfy the rule.

**Results on accuracy and rule teaching efficacy:** Fig. 3 shows how the control parameter affects the task-based metric and the rule-based metric, respectively. The additionally-incorporated rule in DEEPCTRL is hugely beneficial for obtaining more accurate predictions of the next state, $\hat{\boldsymbol{x}}_{t+1}$. Compared to TASKONLY, DEEPCTRL reduces the prediction MAE significantly – the parameters are driven to learn better representations with the domain knowledge constraints. Moreover, DEEPCTRL provides much higher verification ratio than TASKONLY and the resultant energy is not only verified, but also more stable (close to that of TASK&RULE) (Fig. 3c). It demonstrates that DEEPCTRL is able to incorporate the domain knowledge into a data-driven model, providing reliable and robust predictions. DEEPCTRL allows to control model's behavior by adjusting $\alpha$ without retraining: when $\alpha$ is close to 0 in Fig. 3a and 3b, DEEPCTRL is in the *task only* region where its rule-based metric is degraded; by increasing $\alpha$, the model's behavior is more dominated by the rule-based embedding $\boldsymbol{z}_r$ – i.e. although the prediction error is increased, the output is more likely to follow the rule. However, DEEPCTRL is still better than TASKONLY in both metrics regardless of the value of $\alpha$.

**Comparison to baselines:** Fig. 3(a, b) shows a comparison of DEEPCTRL to the baselines of training with a rule-based constraint as a fixed regularization term. We test different $\lambda \in \{0.01, 0.1, 1.0\}$ and all show that the additive regularization (Eq. 1) is helpful for both aspects. The highest $\lambda$ provides the highest verification ratio, however, the prediction error is slightly worse than that of $\lambda = 0.1$. We find that the lowest prediction error of the fixed baseline is comparable to (or even larger than) that of DEEPCTRL, but the highest verification ratio of the fixed baseline is still lower than that of DEEPCTRL. The computed energy values from the fixed baseline are also similar to those from DEEPCTRL. In addition, we consider the benchmark of imposing the rule-constraint with LDF and demonstrate two results where its hyperparameters are chosen by the lowest MAE (LDF-MAE) and highest rule verification ratio (LDF-RATIO) on validation set, respectively. We note that LDF does not allow the capability of flexibly changing the rule strength at inference as DEEPCTRL, so it needs to be retrained for different hyperparameters to find such operation points. LDF-MAE yields higher MAE and lower rule verification ratio compared to others on the test set, showing lack of generalizability of the learned rule behavior. On the other hand, LDF-RATIO provides lower MAE than others. However, only 50% of outputs follow the rule which significantly lowers the reliability of the method. Overall, these results demonstrate that DEEPCTRL is competitive in MAE compared to fixed methods like LDF, while providing the flexibility of adjusting the rule strength to operate at a more favorable point in terms of rule verification ratio, and enabling the extra capabilities presented next.

**Scale of the objectives:** It is important that the scales of $\mathcal{L}_{rule}$ and $\mathcal{L}_{task}$ are balanced in order to avoid all learnable parameters becoming dominated by one of the objective. With the proposed scale parameter $\rho$, the combined loss $\mathcal{L}$ has to be of a similar scale for the two extremes ($\alpha \to 0$ and $\alpha \to 1$). Fig. 3a shows that this scaling enables a more U-shape curve, and the verification ratio when $\alpha \to 0$ is close to that of TASKONLY in Fig. 3b. In other words, DEEPCTRL is close to TASKONLY as $\alpha \to 0$ and close to RULEONLY as $\alpha \to 1.0$, which is a model trained with a fixed $\alpha = 1.0$. It is not necessary to search the scale parameter before training; instead, it is fixed at the beginning of training with the initial loss values of $\mathcal{L}_{rule}$ and $\mathcal{L}_{task}$.

**Optimizing rule strength on validation set:** Users can run inference with DEEPCTRL with any value for $\alpha$ without retraining. In Fig. 3 we consider the scenario where users pick an optimal $\alpha$ based on a target verification ratio. For a goal of $> 90\%$ verification ratio on the validation set, $\alpha > 0.6$ is needed, and this guarantees a verification ratio of $80.2\%$ on the test set. On the other hand, if we consider optimization of $\lambda$ of the fixed objective function ('Task & Rule') benchmark for $> 90\%$ validation verification ratio, we observe a test verification ratio of $71.6\%$, which is $8.6\%$ lower than DEEPCTRL. In addition, the verification ratio on the validation set can be used to determine the rule strength that minimizes error. On the validation set, minimum MAE occurs when $\alpha = 0.2$, which corresponds to a rule verification ratio around 60 %. If we search for the $\alpha$ that satisfies the same rule verification ratio of 60 % on the test set, we observe that $\alpha = 0.4$ is optimal, which also gives

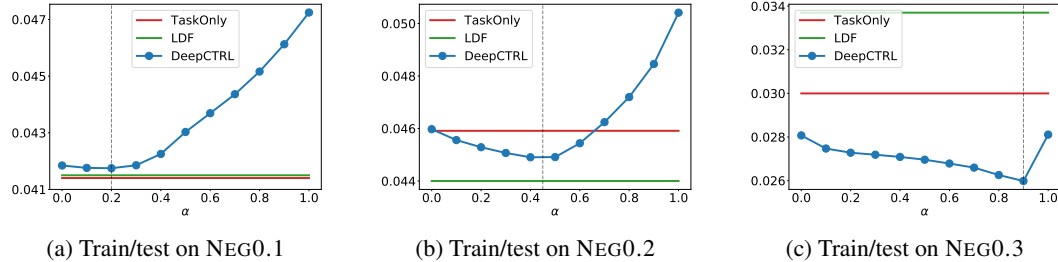

(a) Train/test on NEG0.1      (b) Train/test on NEG0.2      (c) Train/test on NEG0.3

Figure 4: Candidate rule testing. Sales prediction error (MAE) across three groups with different correlation coefficients between $\Delta$SALES and $\Delta$PRICES. The dashed lines point the optimal $\alpha$ providing the lowest error.

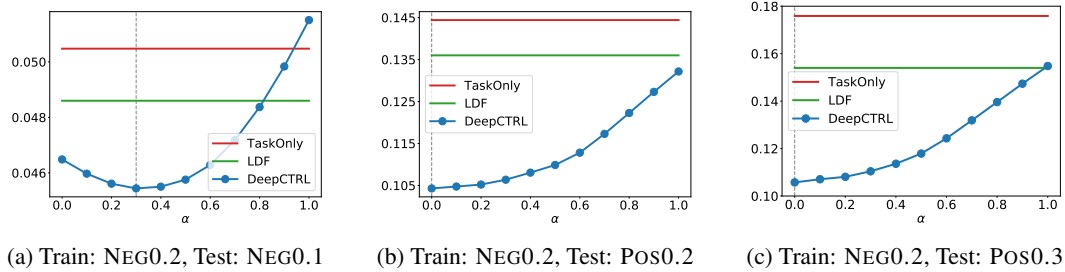

(a) Train: NEG0.2, Test: NEG0.1     (b) Train: NEG0.2, Test: POS0.2     (c) Train: NEG0.2, Test: POS0.3

Figure 5: Candidate model testing. Sales prediction error (MAE) for three groups from a model trained on NEG0.2. Note that the correlation coefficients in the target groups are different to that of the source group. The dashed lines point the optimal $\alpha$ providing the lowest error.

the lowest MAE. In other words, the rule verification ratio is a robust unsupervised model selection proxy for DEEPCTRL, underlining the generalizability of the learned representations.

## 4.2 Examining Candidate Rules

DEEPCTRL allows 'hypothesis testing' for rules. In many applications, rules are not scientific principles but rather come from insights, as in economical or sociological sciences. We do not always want rules to dominate data-driven learning as they may hurt accuracy when they are not valid.

**Dataset, task and the rule:** We focus on the task of sales forecasting of retail goods on the M5 dataset [1]. While the original task is forecasting daily sales across different goods at every store, we change the task to be forecasting weekly sales since the prices of items are updated weekly. For this task, we consider the economics principle as the rule [6]: *price-difference and sales-difference should have a negative correlation coefficient*: $r = \frac{\Delta \text{SALES}}{\Delta \text{PRICES}} < 0.0$. This inductive bias is incorporated using the perturbation-based objective described in Sec. 3. The positive perturbation is applied to *price* of an input item, and $\mathcal{L}_{rule}$ is a function of the perturbed output $\hat{y}_p$ and the original output $\hat{y}$: $\mathcal{L}_{rule}(\hat{y}, \hat{y}_p) = \text{ReLU}(\hat{y}_p - \hat{y})$. Unlike the previous task, the rule of this task is *soft* – not all items are known to follow the rule and their correlation coefficients are different.

**Experimental setting:** We split items in three groups: (1) NEG0.1 items with a negative price-sales correlation ($r < -0.1$), (2) NEG0.2 items with negative price-sales correlation ($r < -0.2$), and (3) NEG0.3 items with negative correlation ($r < -0.3$). We train TASKONLY, LDF, and DEEPCTRL on each group to examine how beneficial the incorporated rule is. Additionally, we examine candidate models by applying a model trained on NEG0.2 to unseen groups: (1) NEG0.1, (2) POS0.2, and (3) POS0.3 where the rule is weak in the latter two groups.

**Candidate rule testing:** Fig. 4 shows the weekly sales prediction results. Compared to TASKONLY, DEEPCTRL obtains lower MAE on NEG0.2 and NEG0.3, but TASKONLY outperforms DEEPCTRL when the items have weaker correlations between $\Delta$SALES and $\Delta$PRICES. This is reasonable since the rule we impose is more dominant in NEG0.2 and NEG0.3, however, it is less informative for

[1] https://www.kaggle.com/c/m5-forecasting-accuracy/

items in Neg0.1. While LDF provides lower MAE on Neg0.2, its performance is significantly degraded on Neg0.3 due to an unstable dual learning process. Furthermore, the rule-dependency can be discovered via the value of $\alpha$ providing the lowest error. Fig. 4 clearly demonstrates that the optimal $\alpha$ is shifted to larger values as the correlation gets stronger. In other words, as items have stronger correlations between $\Delta$PRICES and $\Delta$SALES, the corresponding rule is more beneficial, and thus the rule-dependency is higher. The post-hoc controllability of DEEPCTRL enables users to examine how much the candidate rules are informative for the data.

**Candidate model testing:** The hypothesis testing use case can be extended to model testing to evaluate appropriateness of a model for the target distribution. Fig. 5 shows models testing capability, where items in each domain have different price-sales correlations. Overall, DEEPCTRL is superior to other methods like LDF as it learns the task and rule representations in disentangled ways, so that the negative impact of a rule is more minimal via a lower when the rule is not helping for the task. There are notable points from the experimental results. First, Fig. 5a shows that TASKONLY, LDF, and DEEPCTRL can be applicable to target domain (NEG0.1) without significant performance degradation. Compared to Fig. 4a, MAE from TASKONLY is increased from 0.041 to 0.05 because TASKONLY is optimized on NEG0.2. While DEEPCTRL is also degraded (0.042 to 0.045), the gap is much smaller than that of TASKONLY as we can optimize DEEPCTRL by tuning $\alpha$. Second, compared to Fig. 4b, we see that the optimal $\alpha$ is decreased to 0.3 from 0.45, showing that rule-dependency becomes weaker compared to source domain. Third, if the model is applied to (POS0.2 and POS0.3) where the imposed rule for the source domain is not informative at all, not only the prediction quality is significantly degraded, but also the rule-dependency is minimized (the optimal $\alpha$ is 0). This indicates the benefit of controllability to examine how much the existing rules or trained models are appropriate for given test samples.

## 4.3 Adapting to Distribution Shifts using the Rule Strength

There may be shared rules across multiple datasets for the same task – e.g. the energy conservation law must be valid for pendulums of different lengths or counts. On the other hand, some rules may be valid with different strengths among different subsets of the data – e.g. in disease prediction, the likelihood of cardiovascular disease with higher blood pressure increases more for older patients than younger patients. When the task is shared but data distribution and the validity of the rule differ among different datasets, DEEPCTRL is useful for adapting to such distribution shifts by controlling $\alpha$, avoiding then need for fine-tuning or training from scratch.

**Dataset, task and the rule:** We evaluate this aspect with a cardiovascular classification task[2]. We consider 70,000 records of patients with half having a cardiovascular disease. The target task is to predict whether the cardiovascular disease is present or not based on 11 continuous and categorical features. Given that higher systolic blood pressure is known to be strongly associated with the cardiovascular disease [19], we consider the rule "$\hat{y}_{p,i} > \hat{y}_i$ if $x_{p,i}^{press} > x_i^{press}$", where $x_i^{press}$ is systolic blood pressure of the $i$-th patient, and $\hat{y}_i$ is the probability of having the disease. The subscript $p$ denotes perturbations to the input and output. Fitting a decision tree, we discover that if blood pressure is higher than 129.5, more than 70% patients have the disease. Based on this information, we split the patients into two groups: (1) $\{i : \{x_i^{press} < 129.5 \cap y_i = 1\} \cup \{x_i^{press} \geq 129.5 \cap y_i = 0\}\}$ (called UNUSUAL) and (2) $\{i : \{x_i^{press} < 129.5 \cap y_i = 0\} \cup \{x_i^{press} \geq 129.5 \cap y_i = 1\}\}$ (called USUAL). The majority of patients in the UNUSUAL group likely to have the disease even though their blood pressure is relatively lower, and vice versa. In other words, the rule is not very helpful since it imposes a higher risk for higher blood pressure. To evaluate the performance, we create different target datasets by mixing the patients from the two groups with different ratios (See Appendix).

**Target distribution performance:** We first train a model on the dataset from the source domain and then apply the trained model to a dataset from the target domain without any additional training. Fig. 6a shows that TASKONLY outperforms DEEPCTRL regardless of the value of $\alpha$. This is expected as the source data do not always follow the rule and thus, incorporating the rule is not helpful or even harmful. Note that the error monotonically increases as $\alpha \to 1$ as the unnecessary (and sometimes harmful) inductive bias gets involved more. When a trained model is transferred to the target domain, the error is higher due to the difference in data distributions. However, we show that the error can be reduced by controlling $\alpha$. For TARGET 1 where the majority of patients are from USUAL, as $\alpha$ is increased, the rule-based representation, which is helpful now, has more weight and the resultant

---

[2]https://www.kaggle.com/sulianova/cardiovascular-disease-dataset

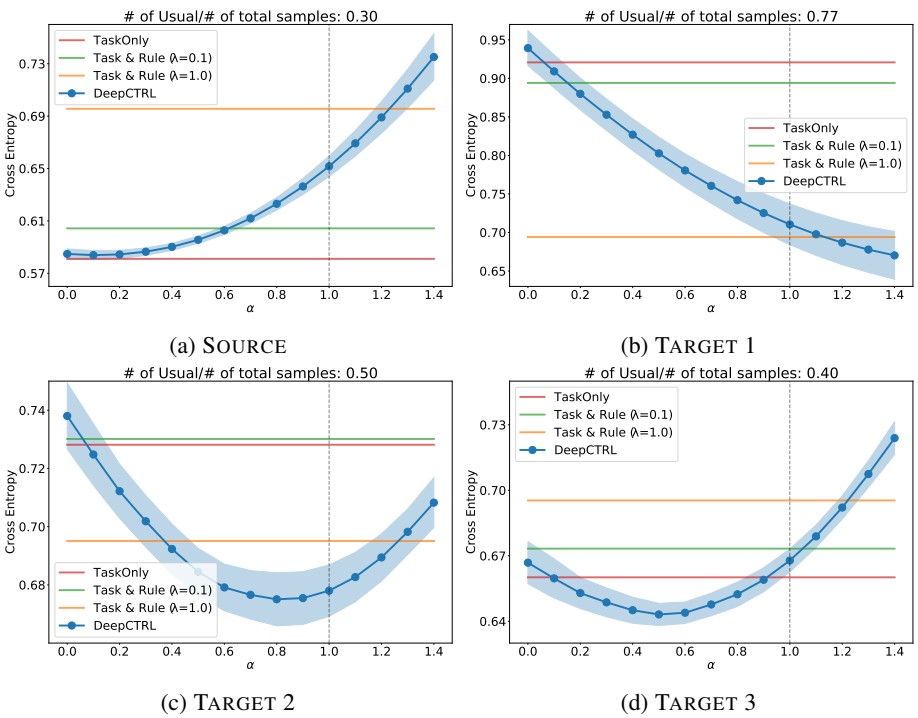

Figure 6: Test cross entropy vs. rule strength for target datasets with varying USUAL/UNUSUAL ratio.

error decreases monotonically. When the ratio of patients from USUAL is decreased, the optimal $\alpha$ is an intermediate value between 0 and 1. These demonstrate the capability of DEEPCTRL to adapt the trained model via $\alpha$ towards an optimal behavior if the rule is beneficial for the target domain. Moreover, we can reversely interpret the amount of rule dependency for a target dataset. As the optimal $\alpha$ is close to 1.0 for TARGET 1, we expect that the rule is valid for most of the patients, unlike TARGET 3 which has the optimal $\alpha$ around 0.5.

**Extrapolating the rule strength:** While $\alpha$ is sampled between 0 and 1 during training, the encoder outputs can be continuously changed when $\alpha > 1$ or $\alpha < 0$. Thus, it is expected that the rule dependency should get higher or lower when we set $\alpha > 1$ or $\alpha < 0$, respectively. We find that the rule is effective for TARGET 1 and the error is monotonically decreased until $\alpha = 1$ when we apply the trained model on SOURCE (Fig. 6b). The decreasing-trend of the error is continued as $\alpha$ is increased until 1.4 and it extends the range of $\alpha$ for further controllability. This observation is particularly relevant when it is necessary to increase the rule dependency more and underlines how DEEPCTRL learns rule representations effectively.

## 5 Ablation Studies

**Rule strength prior** $P(\alpha)$**:** To ensure that DEEPCTRL yields desired behavior with a range of $\alpha$ values at inference, we learn to mimic predictions with different $\alpha$ during training to provide the appropriate supervision. We choose the Beta distribution to sample $\alpha$ because via only one parameter, it allows us to sweep between various different behaviors. One straightforward idea is to uniformly simulate all potential inference scenarios during training (i.e. having Beta(1,1)), however, this yields empirically worse results. We propose that overemphasizing on edge cases (very low and very high $\alpha$) is important so that the behavior for edge cases can be learned more robustly, and the interpolation between them would be accurate. Fig. 7a and 7b shows results with different values of Beta($\beta, \beta$). For the pendulum task, $\mathcal{L}_{rule}$ is much larger than $\mathcal{L}_{task}$ and it leads DEEPCTRL getting mostly affected by $\mathcal{L}_{rule}$ on Beta(1.0, 1.0) when $\alpha > 0.1$, hurting the task-specific performance. We observe that there is typically an intermediate $\beta < 1$ optimal for the general behavior of the curves − $\beta = 0.1$ seems to be a reasonable choice across all datasets, and the results are not sensitive when it is slightly higher or lower.

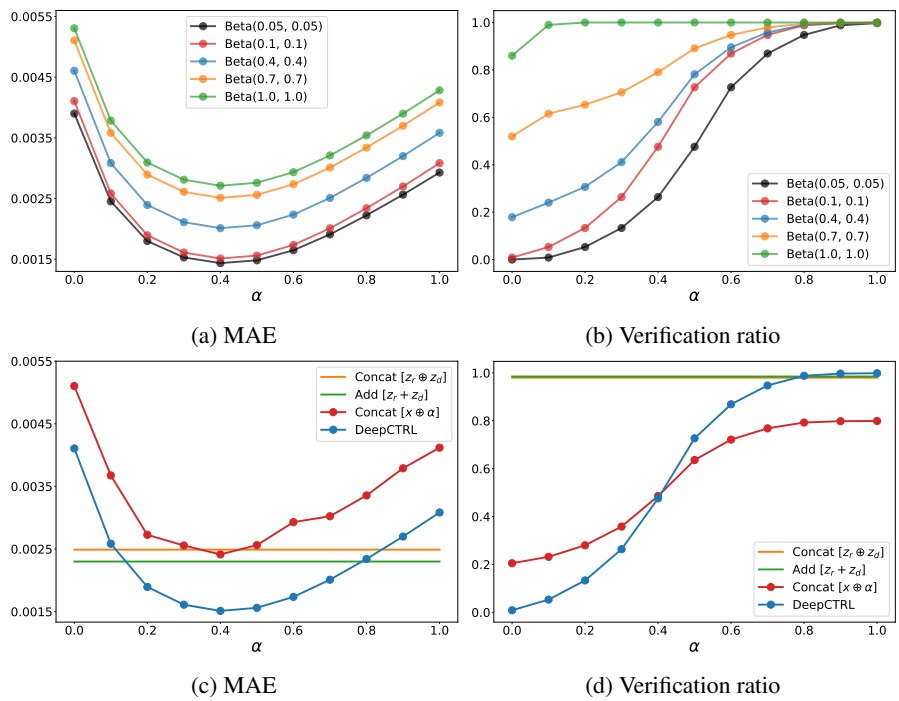

(a) MAE

(b) Verification ratio

(c) MAE

(d) Verification ratio

Figure 7: (a,b) Results on a double pendulum task for various of $\beta$. (c,d) Comparison with two non-coupled representations (CONCAT and ADD) and a concatenated input $(\boldsymbol{x} \oplus \alpha)$.

**Stochastically-coupled representations:** We propose to combine the two representations from the encoders to enable stochastic coupling with corresponding objectives $(\alpha \boldsymbol{z}_r, \alpha \mathcal{L}_{rule}$ and $(1 - \alpha) \boldsymbol{z}_d, (1 - \alpha) \mathcal{L}_{task})$. As an alternative, we consider the coupled representations with two conventional ways of merging: CONCAT $(\boldsymbol{z} = \boldsymbol{z}_r \oplus \boldsymbol{z}_d)$ and ADD $(\boldsymbol{z} = \boldsymbol{z}_r + \boldsymbol{z}_d)$. Fig. 7c and 7d shows that these can provide a very high verification ratio (as $\mathcal{L}_{rule}$ dominates over $\mathcal{L}_{task}$), equivalent to DEEPCTRL at high $\alpha$, but their errors at intermediate $\alpha$ values are higher than DEEPCTRL. The proposed coupling method in DEEPCTRL enables disentangled learning of rule vs. task representations by mixing them with $\alpha$. We also concatenate $\alpha$ as $[x; \alpha]$ instead of relying on the proposed two-passage network architecture. Fig. 7c and 7d shows that concatenating inputs does not improve the task error or verification ratio, although the same amount of information is used. This supports the hypothesis that the coupled separation ($\boldsymbol{z}_r$ and $\boldsymbol{z}_d$) is more desirable for controlling corresponding representations.

## 6    Conclusion and Societal Impact

Learning from rules can be crucial for constructing interpretable, robust, and reliable DNNs. In this paper, we propose DEEPCTRL, a new methodology to incorporate rules into data-learned DNNs. Unlike existing methods, DEEPCTRL enables controllability of rule strength at inference without retraining. We propose a novel perturbation-based rule encoding method to integrate arbitrary rules into meaningful representations. Overall, DEEPCTRL is model architecture, data type and rule type agnostic. We demonstrate three uses cases of DEEPCTRL: improving reliability given known principles, examining candidate rules, and domain adaptation using the rule strength. We leave theoretical proofs for convergence of learning, extensive empirical performance analysis on other datasets, as well as demonstrations of other use cases like robustness, for future work.

DEEPCTRL has many potential benefits in real-world deep learning deployments to improve their accuracy, to increase their reliability, and to enhance human-AI interaction. On the other hand, we also note the capability of DEEPCTRL in encoding rules in effective ways can have undesired outcomes if used with bad intentions to teach unethical biases.

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
