# OpenReview forum: "Controlling Neural Networks with Rule Representations"
_NeurIPS.cc/2021/Conference — NeurIPS 2021 Poster_

### Official Review · Reviewer_hguV · 2021-06-24

**Rating:** 8
**Confidence:** 4

**Summary:**

This work proposed an interesting way of mixing the power of data and rules while enabling the adaptation at the inference time. This paper proposed to use perturbation of the input data points as a novel way of injection rules compared with the continuous relaxation of rules.

**Main Review:**

This learning framework as well as the perturbation-based rule injection looks very interesting and novel to me. First of all, combining rule-based prior knowledge with learned knowledge is an interesting and promising direction for improving neural networks' generalization performance. The proposed perturbation-based rule injection is very novel to me. Most existing works inject rules using their continuous relaxation. The perturbation-based rule injection provides a new way to inject a broader range of rules that are potentially hard to relax.
Second, the idea of controlling the importance weights between data (learned knowledge) and rules is quite novel. Though it is simple from hindsight, it does enable the control of a trained model and provides some interpretability to the model itself (rather than a pure black box).  Moreover, the control of the weights between data and rules can help to solve out-of-sample data points (as shown in their experiments).
Overall I think this paper deserves a score of 8.  I recommend accepting this paper.


One suggestion: I hope the author gives a broader literature survey for methods that try to merge the power of data and rule/symbolic knowledge.

Here are some related works that I know:

Deep Reasoning Networks for Unsupervised Pattern De-mixing with Constraint Reasoning
A semantic loss function for deep learning with symbolic knowledge

Questions:
It looks quite counterintuitive to concatenate the two embeddings z_d and z_r while scaling them by alpha and (1-alpha). People often use this kind of “convex combination of two” on the summation operator rather than concatenation. Did you try if you use z = alpha * z_d + (1-alpha) *z_r?


**Time Spent Reviewing:**

2

---

> ### Author Response · Authors · 2021-08-10
> **Regarding the limitations points and questions**
>
> Thanks for finding our work very interesting and novel!
>
> We have expanded the literature review with the papers you suggested, as well as the ones suggested by Reviewer 9N6z. Please let us know if you think of any other works.
>
> You are totally right that scaling with $\alpha$ and $(1-\alpha)$ is common when considered in convex combinations, as in our loss term. Regarding the embedding combination, we specifically chose concatenation as we would like to assign information to certain dimensions explicitly. Note that the decision block is a low capacity model compared to the encoders, and thus, we don’t want to put the representation disentanglement functionality on it (which would have been the case with addition that would create an undistinguishable combination of the two representations). The proposed concatenation idea is observed to work well canonically across different data types and model architectures, especially when the experiments for superior rule and task based disentanglement are considered.

---

> > ### Comment · Reviewer_hguV · 2021-08-11
> > **Update**
> >
> > Thanks for answering my questions. I'm pretty much satisfied now.
> >
> > To the question of latent embedding: Even if you use the concatenation, we still don't really know what the real semantic is except that half is from data and another half is from the rules. So, practically, there is no much difference between concatenation/addition in terms of explainability. Did you try to use addition instead of concatenation? How does it perform?

---

> > > ### Author Response · Authors · 2021-08-12
> > > **Concatenation and Addition**
> > >
> > > Thanks for your insight about concatenation and addition.
> > >
> > > As you explained, in terms of representational power or information perspective, there is no much difference between concatenation and addition. Ideally, if the model is perfect, it may not lose the information even with the summation, which creates an undistinguishable combination. In this work, we take the concatenation to explicitly split the representations which is a safer way even with imperfect networks.
> > >
> > > To verify that addition and concatenation have similar impact in terms of the information prospective, we conduct additional experiments.
> > >
> > > - Task-based prediction error (MAE) (Fig. 3(a))
> > > | $\alpha$      | 0.0 | 0.1 | 0.2 | 0.3 | 0.4 | 0.5 | 0.6 | 0.7 | 0.8 | 0.9 | 1.0 |
> > > | ----------- | ----------- | ----------- | ----------- | ----------- | ----------- | ----------- | ----------- | ----------- | ----------- | ----------- | ----------- |
> > > | Add (10^-3)   | 4.22 | 2.62 | 1.83 | 1.65 | 1.58 | 1.61 | 1.73 | 2.12 | 2.37 | 2.72 | 3.16|
> > > | Concat (10^-3)  | 4.15 | 2.58 | 1.80 | 1.62 | 1.55 | 1.60 | 1.70 | 2.03 | 2.35 | 2.71 | 3.12|
> > >
> > > - Verification ratio (Fig. 3(b))
> > > | $\alpha$      | 0.0 | 0.1 | 0.2 | 0.3 | 0.4 | 0.5 | 0.6 | 0.7 | 0.8 | 0.9 | 1.0 |
> > > | ----------- | ----------- | ----------- | ----------- | ----------- | ----------- | ----------- | ----------- | ----------- | ----------- | ----------- | ----------- |
> > > | Add   | 0.0 | 0.08 | 0.17 | 0.25 | 0.45 | 0.69 | 0.85 | 0.97 | 0.99 | 1.0 | 1.0 |
> > > | Concat   | 0.0 | 0.05 | 0.15 | 0.25 | 0.48 | 0.72 | 0.85 | 0.96 | 0.99 | 1.0 | 1.0 |
> > >
> > > We find that two methods have shown very similar results.
> > >
> > > We also compare two methods by setting $\rho=1$ in Algorithm 1. As L_{rule} is larger than L_{task}, two encoders are more affected by rule-based objective. Thus, we can see how performance is changed when a rule loss is larger than a task loss.
> > >
> > > - Task-based prediction error (MAE)
> > > | $\alpha$      | 0.0 | 0.1 | 0.2 | 0.3 | 0.4 | 0.5 | 0.6 | 0.7 | 0.8 | 0.9 | 1.0 |
> > > | ----------- | ----------- | ----------- | ----------- | ----------- | ----------- | ----------- | ----------- | ----------- | ----------- | ----------- | ----------- |
> > > | Add (10^-3)   | 0.869 | 0.997 | 1.193 | 1.314 | 1.429 | 1.546 | 1.724 | 1.964 | 2.407 | 2.943 | 3.755 |
> > > | Concat (10^-3)  | 0.856 | 0.974 | 1.202 | 1.460 | 1.689 | 1.840 | 1.848 | 1.857 | 2.036 | 2.410 | 3.281 |
> > >
> > > - Verification ratio
> > > | $\alpha$      | 0.0 | 0.1 | 0.2 | 0.3 | 0.4 | 0.5 | 0.6 | 0.7 | 0.8 | 0.9 | 1.0 |
> > > | ----------- | ----------- | ----------- | ----------- | ----------- | ----------- | ----------- | ----------- | ----------- | ----------- | ----------- | ----------- |
> > > | Add   | 0.64 | 0.90 | 0.97 | 0.99 | 1.0 | 1.0 | 1.0 | 1.0 | 1.0 | 1.0 | 1.0 |
> > > | Concat   | 0.69 | 0.94 | 0.99 | 1.0 | 1.0 | 1.0 | 1.0 | 1.0 | 1.0 | 1.0 | 1.0 |
> > >
> > > While we can see that concatenation is slightly better than addition when a rule loss is larger than a task-specific loss, the difference is not that significant in the double pendulum example.

---

> > > > ### Comment · Reviewer_hguV · 2021-08-14
> > > > **Concatenation and Addition**
> > > >
> > > > Thanks for the additional experiments! I'm very happy now!

---

### Official Review · Reviewer_x3gr · 2021-07-04

**Rating:** 7
**Confidence:** 4

**Summary:**

The paper proposes DeepCTRL, a novel training method to integrate rules into neural networks. DeepCTRL is evaluated through several applications showing how rules can be effectively incorporated and improve performance.

**Limitations And Societal Impact:**

The authors do not seem to explicitly address limitations and societal impact.

**Main Review:**

Overall, the idea proposed is clear and yields strong results.

Fig 4 makes a compelling case that DeepCTRL works and is useful. One question about Fig 4 (especially 4c): it seems to make the case that DeepCTRL does a better job of enforcing the soft rule than LDF. It is surprising then that in 5b and 5c DeepCTRL continues to outperform LDF even when imposing the rule would ostensibly hurt perfromance. Do the authors have an explanation for why this might be?

The results in Fig 3 do not clearly show that DeepCTRL outperforms the LDF-ratio baseline. In this case, LDF-Ratio seems to perform the best, even without following the energy damping rule. Even if its verification ratio is lower, this example does not clearly show that DeepCTRL is outperforming LDF-Ratio.

The ablation studies give reasonable confidence that some of the calls made in DeepCTRL's methodology are sound.

One relatively simple baseline that comes to mind is using a convex combination of two classifiers: the rule-based model and a neural network model. This should still preserve many of the proposed useful properties (e.g. selecting the relative strength of the rule-model at test time). Is there a reason the proposed model is better than this simple baseline and does not need to be compared to it?

**Minor points.**

- grammar in first sentece of abstract is slightly unclear
- fig 1 caption - "DEEPCTRL **provide** a controllable mechanism"
- fig 1 itself is faily confusing - what do the blue arrows represent? Why isn't the physical rule of  monotonicity obeyed by any of the curves (it seems to be even less obeyed for the bottom plot, where the rule strength is ostensibly the highest)?

**Time Spent Reviewing:**

3

---

> ### Author Response · Authors · 2021-08-10
> **Regarding the limitations points and questions**
>
> Thanks for finding our paper clear and our results strong!
>
> ### Why does DeepCTRL outperform LDF in Fig 5b and 5c?
>
> We attribute the outperformance of DeepCTRL in Fig. 5b and 5c to learn the task and rule representations in disentangled ways, so that when the rule is not helpful, via selection of a small $\alpha$, the negative impact of rule representation can be minimized. We have added the following note about this point:
>
> “DeepCTRL is superior to other methods like LDF as it learns the task and rule representations in disentangled ways, so that the negative impact of a rule is more minimal via a lower $\alpha$ when the rule is not helping for the task.”
>
> ### Regarding Fig 3: DeepCTRL vs. LDF-ratio
>
> It is correct that LDF-Ratio achieves slightly lower MAE than DeepCTRL, however, it is at an operation point with low verification ratio. We have removed the expression on the superiority of our model to avoid overclaiming and rephrase to focus on the extra capabilities:
>
> “Overall, these results demonstrate that DeepCTRL is competitive in MAE compared to fixed methods like LDF, while providing the flexibility of adjusting the rule strength to operate at a more favorable point in terms of rule verification ratio, and enabling the extra capabilities presented next.”
>
>
> ### A convex combination of two simple classifiers
>
> Considering the convex combination of the two classifiers is definitely an interesting idea. We note that the rule encoder model is supervised with the rule-based objective, so to map the representation to the task-based decision (e.g. class probabilities), we still need a classifier block that can do nonlinear mapping of the representation. Thus, with simple convex combination, which in a way replaces the nonlinear mapping of the classifier block with simple linear scaling, the perturbation from rule representation would hurt the classification accuracy significantly.
>
>
> ### Unclear the first sentence in abstract
>
> We have clarified the first Abstract sentence as:
>
> “We propose a novel training method that integrates rules into deep learning, in a way the strengths of the rules are controllable at inference.”
>
> ### Unclear Fig 1.
>
> We have fixed the grammar error in Fig. 1 caption and provided clarifications:
>
> “DeepCTRL (with outputs shown via blue arrows) provides a controllable mechanism that enables the rule dependency to be adjusted at inference time in order to achieve an optimal behavior (middle graph) in regards to accuracy and rule verification ratio. With increased rule strength, DeepCTRL yields an operation point (bottom graph) where $(E_t > \hat{E}_{t+1})$ is satisfied for all time steps.
>
> Note that $E_t$ comes from the red curve (ground truth) and $\hat{E}_{t+1}$ is from prediction.
>
> With added noise in the data, the model provides fluctuating output (as described in blue curves) and it often violates the rule (greater than red curve). By increasing the rule strength, we guarantee that $\hat{E}_{t+1}$ (blue curve) is always under the $E_t$. Furthermore, the fluctuation is also minimized when we find a proper strength (middle graph).
>
> ### Societal Impact
>
> Thanks for noting the potential Societal Impact. We agree the paper could benefit from a discussion on this and have added the following:
>
> “DeepCTRL has many potential benefits in real-world deep learning deployments to improve their accuracy, to increase their reliability, and to enhance human-AI interaction. On the other hand, we also note the capability of DeepCTRL in encoding rules in effective ways can have undesired outcomes if used with bad intentions to teach unethical biases.”

---

> > ### Comment · Reviewer_x3gr · 2021-08-12
> > **Reviewer response**
> >
> > Thanks for answering my questions - all the responses seem reasonable, although it still seems to be that the convex combination of 2 classifiers would be a reasonable baseline, and it is not obvious that "the perturbation from rule representation would hurt the classification accuracy significantly."

---

> > > ### Author Response · Authors · 2021-08-12
> > > **A convex combination of two simple classifiers**
> > >
> > > Thanks for pointing out the possibility of a convex combination of two simple classifiers.
> > >
> > > We conducted additional experiments to verify how two simple classifiers behave when the two classifiers are convex-combined.
> > > For the healthcare experiment (Sec 4.3), we define two classifiers: (1) rule-classifier (RC) and (2) data-classifier (DC).
> > >
> > > RC is supervisedly trained by the perturbation-based objective defined in Sec. 4.3.
> > > Given input set $x$ and perturbed input set $x_p$, RC returns $\hat{y}$ and $\hat{y}_p$, respectively. Then, we impose the rule constraint ($\hat{y}_p>\hat{y}$ if $x_p>x$, see line 278) to both output to update learnable parameters in RC.
> > >
> > > DC is also supervisedly trained by the task-based objective, which is cross-entropy.
> > >
> > > Once RC and DC are trained, we combine the two classifiers as follow:
> > > Given input $x$, we have output from RC $(\hat{y}_{RC})$.
> > >
> > > Similarly, we have output from DC $(\hat{y}_{DC})$.
> > >
> > > Then, we combine RC and DC (called Two-cls), and the final output is
> > >
> > > $\hat{y}=\alpha\hat{y}_{RC} + (1-\alpha)\hat{y}{DC}$
> > >
> > > where $\alpha$ is sampled from the same distribution used in Sec. 4.3.
> > >
> > > We test Two-Cls on the Source split (Fig 6.a).
> > >
> > > | $\alpha$      | 0.0 | 0.1 | 0.2 | 0.3 | 0.4 | 0.5 | 0.6 | 0.7 | 0.8 | 0.9 | 1.0 |
> > > | ----------- | ----------- | ----------- | ----------- | ----------- | ----------- | ----------- | ----------- | ----------- | ----------- | ----------- | ----------- |
> > > | Two-Cls   | 0.585 | 0.586 | 0.595 | 0.604 | 0.609 | 0.624 | 0.640 | 0.660 | 0.680 | 0.706 | 0.730 |
> > > | DeepCTRL   | 0.583 | 0.582 | 0.583 | 0.585 | 0.588 | 0.592 | 0.603 | 0.612 | 0.620 | 0.632 | 0.650 |
> > >
> > > When $\alpha$ is close to 0 where data-driven model is dominant, Two-Cls and DeepCTRL provide similar results. This is expected result as both Two-Cls and DeepCTRL become one data-driven classifier.
> > > However, as $\alpha$ increases, the cross entropy loss of Two-Cls is rapidly increased. This behavior comes from the limited RC. Since RC is solely trained by the perturbation-based constraint which is correlation between a particular input feature (systolic blood pressure) and ouput (risk), it is easy to train RC to satisfy the rule but the learned parameters won't utilize other features which can be helpful.
> > > In other words, as $\alpha$ increases, Two-Cls becomes more and more RC which only depends on the systolic blood pressure, and thus, the classifier's expressive power is limited and the cross-entropy is more increased.
> > >
> > > In DeepCTRL, the rule-encoder is affected by task-based objective (Fig. 2) so that it can learn both rule-specific and task-specific representations.

---

### Official Review · Reviewer_xjoF · 2021-07-15

**Rating:** 6
**Confidence:** 5

**Summary:**

This paper proposes to integrate a combination of rule learning and data learning in the deep neural networks. The core motivation is to introduce prior knowledge and/or other learning methods and learning how to "balance" the pure data-driven part of the neural network with its rule learning component. The final neural network has two components: the rule-based encoder and the data-driven encoder that are combined by a parametrized convex combination. The loss function has also a counterpart that captures the traditional loss function with penalizations on the rule violations. The second part is similar to the Lagrangian approach to training neural networks subject to constraints that has been used successfully in prior (mostly cited) work. The novelty is the introduction of the rule encoder inside the neural network, as examplified by the third application. The paper also proposes an interesting perturbation approach to capture some of the rule behavior.

**Limitations And Societal Impact:**

The paper mentions some of its limitations at the end. The paper should mention that the rule-based encoder may codify existing biases in some possible applications.


**Main Review:**

It is an interesting paper giving an interesting direction. Some of the work described is a direct consequence of prior work on incorporating constraints in the objective function. But the paper generalizes this with two ideas
    -- the idea of integrating rule learning directly in the neural networks
    -- the idea of local perturbation, which is quite nice
These ideas are an interesting avenue for research in a variety of applications.

There are some weaknesses in the paper. The paper lists a number of limitations  of prior work
  (1)  the Lagrangian multipliers are fixed;
  (2) the loss function for the "constraint" part must be differentiable
  (3) the components cannot be balanced at inference time.

(1) is not correct and is addressed in prior work. The approach here also suffers from (2). (3) is addressed by changing how to weight the rule and data encoders during inference. It is not clear what the justification is here since the network is trained with a different weight.



**Time Spent Reviewing:**

2 hours

---

> ### Author Response · Authors · 2021-08-10
> **Regarding the limitations points and questions**
>
> Thanks for your valuable feedback on the paper – we're glad you found it interesting!  Please see responses to the questions and comments below.
>
> ### Being more clear on the limitations of prior work
>
> Thank you for pointing these out!  Please see our responses below.  We have clarified these points in the paper to improve the clarity of the limitations of the past work and how our paper differentiates.
>
> ### Fixed Lagrangian multipliers
>
> In LDF, Lagrangian multipliers are not fixed but updated during training in dual optimization steps. The LDF hyper-parameter mentioned in our paper is the dual learning rate, which can be used to tune the balance between the task and the rule. Unlike DeepCTRL, LDF has no way to tune this task-rule balance at inference time without re-training.
>
> ### The approach also requires differentiable constraints
>
> DeepCTRL method can work with a non-differentiable constraint as well, as we demonstrated in experiments in the paper. The rule based objective, given in Eq. (3) merely depends on the output of the model, with and without perturbations. As long as the model is differentiable, which is the assumption with trainable deep neural networks, the rule-based objective in Eq. (3) would be differentiable, even if the rule itself is not. In our experiments in Sections 4.2 and 4.3, the rules are indeed non-differentiable with respect to the input, as they concern only a subset of the data samples. For example, in Figs. 4 and 5, those subsets are determined based on the correlation coefficients. As the subset selection is non-differentiable, it makes the entire rule non-differentiable.
>
> ### Justification of different $\alpha$
>
> DeepCTRL can operate with different \alpha values which control the rule strength at inference, without retraining. The key to this capability is training with different \alpha values so that the model learns how to operate with a wide range of \alpha without retraining. If trained with fixed \alpha, we would have different versions of the model to be used at inference. Previous work, to our knowledge, does not consider training with different rules strengths for the capability of flexibly operating with different rule strengths at inference without requiring retraining.
>
> ### Societal Impact
>
> Thanks for noting the potential Societal Impact. We agree the paper could benefit from a discussion on this and have added the following:
>
> “DeepCTRL has many potential benefits in real-world deep learning deployments to improve their accuracy, to increase their reliability, and to enhance human-AI interaction. On the other hand, we also note the capability of DeepCTRL in encoding rules in effective ways can have undesired outcomes if used with bad intentions to teach unethical biases.”

---

> > ### Comment · Reviewer_xjoF · 2021-08-11
> > **After rebuttal**
> >
> > Thank you for clarifying the various questions I had. I am not completely sure how you would choose \alpha at inference time. I would also recommend that you acknowlege prior works on Lagrangian methods in learning.
> >
> > Overall, I believe that this paper contains some really interesting and that the authors have answered some of the questions I had.

---

> > > ### Author Response · Authors · 2021-08-12
> > > **Choosing \alpha at inference time**
> > >
> > > Thanks for the positive feedback on the paper and our answers.
> > >
> > > The strategy to pick $\alpha$ at inference time would depend on the application scenario, as exemplified in our experiments. $\alpha$ may be chosen to achieve a certain verification ratio on the validation set (see the subsection “​​Optimizing rule strength on validation set” in Section 4.1), or can be chosen as the rule strength that minimizes error. For the candidate rule and model testing experiments in Section 4.2, analysis over a wide range of $\alpha$ values is needed. For the application of adapting to distribution shifts in Section 4.3, its selection may be guided by domain knowledge on the distribution shift between training and test set. Since adjusting $\alpha$ to control a model doesn’t require additional training or updating learnable parameters, a wide range of $\alpha$ can be easily tested. We have clarified this explanation.

---

> > > ### Author Response · Authors · 2021-08-12
> > > **Prior works on Lagrangian methods**
> > >
> > > We have added the following works on Lagrangian methods:
> > >
> > > There have been some past works on using Lagrangian dual methods for deep learning. Nandwani et al. [1] applies Lagrangian dual formulation (LDF) on part-of-speech (POS) tagging constraints to improve various NLP tasks such as semantic role labeling and named entity recognition (NER). Fioretto et al. [2] and Rong et al. [4] both propose to use LDF on rules that came from prior knowledge or scientific theories. Tran et al. [3] also applied LDF in the differentiable privacy domain by formulating fairness constraints as the dual problem.
> > >
> > > ### References
> > >
> > > [1] Yatin Nandwani, Abhishek Pathak, Mausam and Parag Singla, A Primal-Dual Formulation for Deep Learning with Constraints. NeurIPS 2019
> > >
> > > [2] Ferdinando Fioretto, Pascal Van Hentenryck, Terrence WK Mak, Cuong Tran, Federico Baldo and Michele Lombardi, Lagrangian Duality for Constrained Deep Learning. ECML PKDD 2020
> > >
> > > [3] Cuong Tran, Ferdinando Fioretto and Pascal Van Hentenryck, Differentially Private and Fair Deep Learning: A Lagrangian Dual Approach. AAAI 2021.
> > >
> > > [4] Miao Rong, Dongxiao Zhang and Nanzhe Wang, A Lagrangian Dual-based Theory-guided Deep Neural Network. arXiv 2008.10159

---

### Official Review · Reviewer_9N6z · 2021-07-16

**Rating:** 6
**Confidence:** 4

**Summary:**

This paper proposes DeepCTRL, a method for infusing domain knowledge in the form of rules into the representation learning models. The method independently encodes the input instance and the supplied rule and couples the input and rule encodings where the coupling is controlled by a parameter \alpha. Training the model involves a task-specific loss function and a “rule-satisfaction” loss function which are again coupled by the same parameter \alpha. Experiments on three tasks related to Physics, Forecasting, and Health domain demonstrate several benefits of the proposed approach which include (i) ability to specify the domain knowledge, (ii) More control on coupling the information from rule and data during the test time (iii) More interpretability.
However, the experiments in this paper seem to be very simplistic and lack exhaustive comparisons with the prior work and thus it’s difficult to judge the practical utility of the proposed method (E.g. all the experiments involve just one rule).

**Limitations And Societal Impact:**

Yes

**Main Review:**

Pros:
- This paper considers an important problem of injecting domain knowledge into representation learning models.
- The proposed method allows easy adaptation of rules to new domains and more controllability in the information coming from rules vs data, during inference.
- Experiments cover multiple domains like Physics, Health and Forecasting (however the datasets seem to be very simplistic)

Limitations:
- Lack of comparisons with the relevant baselines: In section 4.1, I believe [1] is an important baseline, which also tries to combine domain knowledge along with data for learning better models for predicting damped pendulum dynamics. The is a large body of work around learning from rules e.g. [2] and [3]. How does this work compares with these methods? I think the paper should include some more baselines since there are a lot of papers around learning from rules.
- All the experiments deal with just one rule and the datasets seem to be very simplistic. How well does this method do when we have significantly more rules?

Questions:
- In Figure 4a, why is Task-Only better than DeepCTRL for \alpha=0? Should both the methods differ significantly for \alpha=0?

References:

[1] Augmenting Physical Models with Deep Networks for Complex Dynamics Forecasting, ICLR 2021

[2] Learning from Rules Generalizing Labeled Exemplars, ICLR 2020

[3] Snorkel: Fast training set generation for information extraction, VLDB



**Time Spent Reviewing:**

5

---

> ### Author Response · Authors · 2021-08-10
> **Regarding the limitations points and questions**
>
> Thanks for your valuable feedback on our paper and for valuing how it tackles an important problem and how the experiments cover a wide set of domains.  Please see our responses to the specific comments below.
>
>
> ### Lack of baseline
> We agree that [1] (APHYNITY) is an important baseline to include, thanks for pointing that out. In addition, we have included the two additional references you suggested.
>
> We have conducted double pendulum experiments with [1] (called APHYNITY), and observed the following results.
>
> - Task-based prediction error (MAE) (Fig. 3(a))
> |APHYNITY $(\lambda_0,\tau_1,\tau_2)$  | MAE |
> | ---- | ---- |
> |  $(1.0,1.0,0.1)$ |  0.043  |
>
> | $\alpha$      | 0.0 | 0.1 | 0.2 | 0.3 | 0.4 | 0.5 | 0.6 | 0.7 | 0.8 | 0.9 | 1.0 |
> | ----------- | ----------- | ----------- | ----------- | ----------- | ----------- | ----------- | ----------- | ----------- | ----------- | ----------- | ----------- |
> | DeepCTRL   | 0.0041 | 0.0025 | 0.0018 | 0.0016 | 0.0015 | **0.00155** | 0.0017 | 0.0020 | 0.00235 | 0.0027 | 0.0031|
>
> - Verification ratio (Fig. 3(b))
> |APHYNITY $(\lambda_0,\tau_1,\tau_2)$  | Verification ratio |
> | ---- | ---- |
> |  $(1.0,1.0,0.1)$ |  0.50  |
>
> | $\alpha$      | 0.0 | 0.1 | 0.2 | 0.3 | 0.4 | 0.5 | 0.6 | 0.7 | 0.8 | 0.9 | 1.0 |
> | ----------- | ----------- | ----------- | ----------- | ----------- | ----------- | ----------- | ----------- | ----------- | ----------- | ----------- | ----------- |
> | DeepCTRL   | 0.0 | 0.05 | 0.15 | 0.25 | 0.48 | **0.72** | 0.85 | 0.96 | 0.99 | 1.0 | 1.0 |
>
> Overall, the baseline (APHYNITY) is similar to LDF-MAE for the prediction error and LDF-Ratio for the verification ratio (See Fig. 3 a,b). We can find $\alpha$ (e.g., 0.5) which provides smaller prediction error and larger verification ratio from DeepCTRL compared to APHYNITY.
>
> The goal of DeepCTRL is fundamentally different to that of APHYNITY. APHYNITY is designed to incorporate *incomplete* physical dynamics driven by an equation $\frac{dX_t}{dt}=F(X_t)$, which is directly related to a given task. Thus, the known constraint is supposed to be derived from the first order derivation. However, what we proposed focuses on more general rules as described in section 4.2 and 4.3 (e.g., non differentiable rules).
>
> Moreover, we enable us to control the weight of prior knowledge at inference. In the APHYNITY paper, the hyperparameters $(\lambda,\tau)$ are fixed or adjusted during training, however, there is no option to further adjust them to make a trained model adaptable to different prior knowledge dependency.
>
> ### How to handle more rules?
>
> Our experiments aim to illustrate the capabilities of DeepCTRL in the most clear way, so we have focused on easy-to-understand tasks and experiments that correspond to important real-world problems. DeepCTRL can be extended to multiple rules, essentially in two ways. The first way is combining multiple rules into a single “rule” for the purpose of training DeepCTRL. For example, the rules $x>3 \rightarrow y=1$ and $x<-10 \rightarrow y=-1$ can be combined into a single rule, and can be fed into DeepCTRL training algorithm as it merely depends on running inference via the rule-based model. This is more scalable but less flexible for rule controls. The second way would be combining multiple rules via multiple rule encoders and multiple rule strengths. (Multiple rule strengths during training can be sampled from Dirichlet distribution.) This path is particularly desirable if we aim to control the impact of the individual rules separately at inference. We have expanded the explanations on this point and will add experiments on multiple rules to the paper (and will post here if they are ready before discussion closes).
>
> ### Why is Task-Only better than DeepCTRL for $\alpha=0$?
>
> In Fig. 4a, DeepCTRL is trained with a rule which is weak and not very beneficial to the task. Although DeepCTRL aims to decouple the rule representation so that its impact can be minimized when $\alpha=0$ is set at inference, the gap with the task-only indicates the shortcoming of the DeepCTRL in decoupling the rule representation, i.e. the less-beneficial rule is still negatively affecting the learned representations. An ideal method for rule representation learning would help reduce the gap further. We have added this as a limitation.

---

> > ### Comment · Reviewer_9N6z · 2021-08-28
> > **After rebuttal**
> >
> > Thanks for including additional experiments including comparisons with APHYNITY.
> > I'll update my rating accordingly.

---

### Decision · Program_Chairs · 2021-09-27

**Decision:**

Accept (Poster)

**Comment:**

This paper considers deep learning models for applications where rules are important (e.g. physics). The idea is to construct a "rule encoder" to incorporate user-specified rules. Specifically, this integration is done by perturbing the observations and enforce some user-specified rules to "regulate" between the model's outputs on the original & perturbed inputs.

Reviewers conclude that the proposed approach is novel and the machine learning tasks considered in the paper are important. Initially they have concerns on missing comparisons with other approaches. This is partly addressed by further experimental results provided by the author feedback.

Personally I think the paper can be an interesting presentation to the NeurIPS community. I find this paper also connected to existing works related to constrained optimisation/control and adversarial training: these works also consider regularising networks to satisfy some constraints and/or make sure the network returns similar results on perturbed inputs. It would be useful to discuss them in the camera ready.